# Synthesis of Substituted 1,2-Dihydroisoquinolines by Palladium-Catalyzed Cascade Cyclization–Coupling of Trisubstituted Allenamides with Arylboronic Acids

**DOI:** 10.3390/molecules29122917

**Published:** 2024-06-19

**Authors:** Masahiro Yoshida, Ryunosuke Imaji, Shinya Shiomi

**Affiliations:** Faculty of Pharmaceutical Sciences, Tokushima Bunri University, 180 Nishihamabouji, Yamashiro-cho, Tokushima 770-8514, Japan; s193005@stu.bunri-u.ac.jp (R.I.); shinya_shiomi@ph.bunri-u.ac.jp (S.S.)

**Keywords:** allenamide, cascade reaction, palladium catalyst, cyclization, coupling, organoboronic compound, 1,2-dihydroisoquinoline

## Abstract

1,2-Dihydroisoquinolines are important compounds due to their biological and medicinal activities, and numerous approaches to their synthesis have been reported. Recently, we reported a facile synthesis of trisubstituted allenamides via *N*-acetylation followed by DBU-promoted isomerization, where various substituted allenamides were conveniently synthesized from readily available propargylamines with high efficiency. In light of this research background, we focused on the utility of this methodology for the synthesis of substituted 1,2-dihydroisoquinolines. In this study, a palladium-catalyzed cascade cyclization–coupling of trisubstituted allenamides containing a bromoaryl moiety with arylboronic acids is described. When *N*-acetyl diphenyl-substituted trisubstituted allenamide and phenylboronic acid were treated with 10 mol% of Pd(OAc)_2_, 20 mol% of P(*o*-tolyl)_3_, and 5 equivalents of NaOH in dioxane/H_2_O (4/1) at 80 °C, the reaction proceeded to afford a substituted 1,2-dihydroisoquinoline. The reaction proceeded via intramolecular cyclization, followed by transmetallation with the arylboronic acid of the resulting allylpalladium intermediate. A variety of highly substituted 1,2-dihydroisoquinolines were concisely obtained using this methodology because the allenamides, as reaction substrates, were prepared from readily available propargylamines in one step.

## 1. Introduction

Isoquinolines and their derivatives, especially 1,2-dihydroisoquinolines, are among the important structure classes of chemical substances. A wide variety of natural products and biologically active pharmacophores have been reported [1,2,3,4,5,6,7,8], such as acetoneberberine IK-2 (**I**) [5], cribrostatin 4 (**II**) [6], *N*-carboxymethyl compound **III** for a carrier for brain-specific delivery [7], and nitro-substituted 1,2-dihydroisoquinoline **IV** as an HIV-1 inhibitor [8]. For this reason, numerous approaches to the synthesis of 1,2-dihydroisoquinolines have been developed (Figure 1) [9,10,11,12,13,14,15,16,17,18,19,20,21,22].

Allenamides are powerful and versatile synthetic building blocks in organic synthesis, extensively utilized as reaction substrates to produce a variety of synthetically useful organic molecules [23,24]. Among them, palladium-catalyzed cascade cyclization of ortho-haloaryl-substituted allenamides provides efficient approaches for the synthesis of *N*-heterocyclic compounds (Figure 1, eq 1) [25,26,27,28,29,30,31,32,33,34,35,36,37]. The key intermediate in this strategy is the π-allylpalladium species, which is generated by an oxidative addition and allene insertion sequence. Diverse nucleophiles or organic main group element compounds are applied to undergo subsequent allylic substitution reactions, yielding a variety of substituted heterocycles. Considerable effort has been devoted to developing methods for the synthesis of various *N*-heterocyclic compounds, but few examples using polysubstituted allenamides have been reported, presumably due to the difficulty of synthesizing polysubstituted allenamides. Recently, we reported a facile synthesis of trisubstituted allenamides via *N*-acetylation followed by DBU-promoted isomerization, where various substituted allenamides were conveniently synthesized from readily available propargylamines with high efficiency (Figure 1, eq. 2) [38]. In light of this research background, we focused on the utility of this methodology for the synthesis of substituted 1,2-dihydroisoquinolines via the palladium-catalyzed cascade cyclization of ortho-haloaryl-substituted allenamides with arylboronic acids. Although similar transformation using arylboronic acids has been previously reported for the synthesis of indole and isoquinolinone derivatives [32,33], no examples using polysubstituted allenamides as the substrates were reported. Herein, we describe a synthesis of highly substituted 1,2-dihydroisoquinolines via the palladium-catalyzed reaction of trisubstituted allenamides containing a *o*-bromoaryl moiety with arylboronic acids (Figure 1, eq 3).

## 2. Results and Discussion

Trisubstituted allenamides for the palladium-catalyzed cascade cyclization were prepared as shown in Figure 2. The three-component reaction of commercially available arylaldehyde, monosubstituted alkyne, and *o*-bromobenzyl amine yielded the propargylamines **1a**–**1e**, which were subjected to reaction with acetic anhydride and DBU, according to our procedure [38], to afford the corresponding trisubstituted allenamides **2a**–**2e** in moderate to good yields, respectively.

The initial attempts for the palladium-catalyzed cascade reaction were carried out using the *N*-acetyl diphenyl-substituted trisubstituted allenamide **2a** with phenylboronic acid (**3a**) (Table 1). When **2a** and **3a** were treated with 5 mol% of Pd(OAc)_2_, 10 mol% of P(*o*-tolyl)_3_, and 5 equivalents of NaOH in dioxane/H_2_O (4/1) at 80 °C [39], the expected reaction proceeded, affording a substituted 1,2-dihydroisoquinoline **4aa** in 78% yield (entry 1). Upon examining the catalyst amounts (entries 2 and 3), it was found that increasing the amounts to 10 mol% of Pd(OAc)_2_ and 20 mol% of P(o-tolyl)_3_ increased the yield to 88% (entry 3). Reaction temperatures were then investigated (entries 4–6). The yield of **4aa** was 86% when the reaction was carried out at 50 °C (entry 4), but a significant decrease in yield was observed when the temperature was lowered to 25 °C (entry 5). The product was obtained in a 76% yield when the reaction temperature was raised to 100 °C (entry 6). The product was produced in a 70% yield when PPh_3_ was used (entry 7), but the yield decreased to 19% when PCy_3_ was used (entry 8). The reactions using bidentate ligands such as DPPE and DPPF also proceeded, giving **4aa** in 47% and 70% yields, respectively (entries 9 and 10).

We next carried out a study on the substrate scope using various arylboronic acids **3b–3i** with **2a** (Figure 2). When 4-methoxyphenylboronic acid (**3b**) was subjected to the reaction, the corresponding 1,2-dihydroisoquinoline **4ab** was obtained in an 82% yield. Arylboronic acids **3c** and **3d,** having dimethoxyphenyl groups, reacted with **2a** to produce the products **4ac** and **4ad** in 98% and 86% yields, respectively. The reaction of **3e**, having a *tert*-butyl group, also proceeded to give the product **4ae** in a 92% yield. The corresponding products **4af** and **4ag** were obtained in good yields from the reactions using 4-chloro- and 4-fluorophenyl boronic acids **3f** and **3g**, respectively. The reaction using 4-acetylphenylboronic acid (**3h**) afforded the product **4ah** in an 85% yield. When 1-naphtylboronic acid (**3i**) was subjected to the reaction, the corresponding 1,2-dihydroisoquinoline **4ai** was produced in a 67% yield.

The reactions using trisubstituted allenamides **2b–2e** with various substituents and phenylboronic acid (**3a**) are summarized in Figure 3. When the substrate **2b**, with a 4-fluorophenyl group at the 1-position, was subjected to the reaction, the corresponding 1,2-dihydroisoquinoline **4ba** was obtained in a 55% yield. The reaction of allenamide **2c**, which had a 1,3-benzodioxole moiety, proceeded to afford the cyclized product **4ca** in an 80% yield. The substrates **2d** and **2e**, containing a 4-fluoro- and 4-methoxyphenyl group at the 3-position, also reacted with **3a** to produce the corresponding substituted products **4da** (**4ag**) and **4ea** (**4ab**) in 66% and 72% yields, respectively.

A plausible mechanism for the cyclization process is shown in Figure 3. The reaction was initiated with the oxidative addition of the aryl bromide moiety of the allenamide **2** to palladium, generating arylpalladium intermediate **A** [25,26,27,28,29,30,31,32,33,34,35,36,37]. This was followed by an intramolecular allene insertion process (**B**) to generate the π-allyl-palladium intermediate **C** [25,26,27,28,29,30,31,32,33,34,35,36,37]. Then, ligand exchange of palladium complex **C** with hydroxide ion occurred, forming hydroxypalladium species **D** [39]. This species underwent transmetallation with the arylborate complex via intermediate **E** to produce the substituted 1,2-dihydroisoquinoline **4**.

## 3. Materials and Methods

All commercially available reagents were used without further purification. All reactions were performed in glassware equipped with a septum under the positive pressure of argon. The reaction mixture was magnetically stirred. Concentration was performed under reduced pressure. The heating experiments were conducted under an oil bath as a heat source. The reactions were monitored by TLC. TLC was performed on pre-coated plates (0.25 mm, silica gel 60F_245_, Merck & Co., Inc., Kenilworth, NJ, USA). Spots were visualized by exposure to UV light or by immersion into a solution of 10% phosphomolybdic acid in ethanol, followed by heating at ca. 200 °C. Column chromatography was performed on silica gel (40–50 μm, Kanto Chemical Co. Lit., Nihonbashi, Tokyo, Japan). NMR spectra were recorded on a Bruker AVANCED III HD-500 (^1^H: 500 MHz, ^13^C: 125 MHz) spectrometer (Bruker Corporation, Billerica, MA, USA) using tetramethylsilane (^1^H NMR at 0.00 ppm) and CDCl_3_ (^13^C NMR at 77.16) as a reference standard. Chemical shifts were reported in ppm. The following abbreviations were used to denote peak multiplicities: s, singlet; d, doublet; t, triplet; q, quartet; quin, quintet; sept, septet; m, multiplet; br, broadened. Mass spectra and high-resolution mass spectra were recorded on JEOL JMS-700 mass spectrometers (double-focusing magnetic sector) (JEOL Ltd., Tokyo, Japan).

### 3.1. General Procedure for the Three-Component Reaction of Arylaldehyde, Alkyne and Amine in Figure 2: Synthesis of Propargylamine ***1a***

To a solution of benzaldehyde (531 mg, 5.00 mmol) in toluene (6 mL), phenylacetylene (766 mg, 7.50 mmol), 2-bromobenzylamine (1.40 g, 7.50 mmol) and CuBr (143 mg, 1.00 mmol) were added at rt under an argon atmosphere. The reaction mixture was then stirred under reflux conditions for 2 h. The reaction was quenched with sat. NH_4_Cl. The aq. mixture was extracted with AcOEt. The organic layer was washed with brine, dried over MgSO_4_, filtered, and concentrated in vacuo to give a crude product, which was purified by means of silica gel column chromatography (hexane/AcOEt = 30/1 to 10/1) to afford propargylamine **1a** (1.71 g, 4.54 mmol, 90%).

### 3.2. N-(2-Bromobenzyl)-1,3-diphenylprop-2-yn-1-amine (***1a***)

Yield 90% (1.71 g, 4.54 mmol); yellow oil; ^1^H-NMR (500 MHz, CDCl_3_): δ 7.62 (d, 2H, *J* = 7.5 Hz), 7.53 (d, 1H, *J* = 7.5 Hz), 7.52–7.46 (m, 4H), 7.38–7.29 (m, 7H), 4.82 (s, 1H), 4.06 (d, 1H, *J* = 6.6 Hz), 4.04 (d, 1H, *J* = 6.6 Hz), 1.92 (s, 1H); ^13^C-NMR (125 MHz, CDCl_3_): δ 140.2, 138.9, 132.9, 131.8 (2C), 130.6, 128.8, 128.6 (2C), 128.3 (2C), 128.2, 127.9, 127.8 (2C), 127.5, 124.3, 123.2, 89.0, 85.9, 54.0, 51.3; HRMS (EI) *m*/*z* calcd for C_22_H_18_NBr [M]^+^ 375.0623, found 375.0626.

### 3.3. N-(2-Bromobenzyl)-1-(4-fluorophenyl)-3-phenylprop-2-yn-1-amine (***1b***)

Yield 99% (2.09 g, 5.31 mmol); yellow oil; ^1^H-NMR (500 MHz, CDCl_3_): δ 7.58 (dd, 1H, *J* = 9.0 and 5.5 Hz), 7.52–7.46 (m, 4H), 7.43 (d, 1H, *J* = 7.5 Hz), 7.31–7.25 (m, 3H), 7.23 (t, 1H, *J* = 7.5 Hz), 7.06–6.99 (m, 3H), 4.77 (s, 1H), 4.03 (d, 1H, *J* = 6.6 Hz), 4.01 (d, 1H, *J* = 6.6 Hz), 1.91 (s, 1H); ^13^C-NMR (125 MHz, CDCl_3_): δ 162.3 (d, *J* = 244 Hz), 138.7, 135.9, 132.8. 131.7 (2C), 130.5, 129.4, 129.3 (2C, d, *J* = 7.9 Hz), 128.7, 128.3 (2C), 127.4, 124.1, 122.9, 115.3 (2C, d, *J* = 21.6 Hz), 88.7, 86.1, 53.2, 51.1; HRMS (EI) *m*/*z* calcd for C_22_H_17_NBrF [M]^+^ 393.0528, found 393.0534.

### 3.4. 1-(Benzo[d][1,3]dioxol-5-yl)-N-(2-bromobenzyl)-3-phenylprop-2-yn-1-amine (***1c***)

Yield 71% (581 mg, 1.38 mmol); colorless oil; ^1^H-NMR (500 MHz, CDCl_3_): δ 7.55 (d, 1H, *J* = 8.0 Hz), 7.49–7.46 (m, 3H), 7.33–7.27 (m, 4H), 7.15–7.11 (m, 2H), 7.07 (d, 1H, *J* = 8.0 Hz), 6.79 (d, 1H, *J* = 8.0 Hz), 5.96 (s, 2H), 4.74 (s, 1H), 4.07 (d, 1H, *J* = 6.6 Hz), 4.03 (d, 1H, *J* = 6.6 Hz), 1.60 (brs, 1H); ^13^C-NMR (125 MHz, CDCl_3_): δ 147.9, 147.3, 138.9, 134.3, 132.9, 131.8 (2C), 130.7, 128.8, 128.4 (2C), 128.3, 128.2, 127.6, 124.3, 123.1, 121.1, 108.4, 108.1, 89.0, 85.9, 53.8, 51.3; HRMS (EI) *m*/*z* calcd for C_23_H_18_NO_2_Br [M]^+^ 419.0521, found 419.0524.

### 3.5. N-(2-Bromobenzyl)-3-(4-fluorophenyl)-1-phenylprop-2-yn-1-amine (***1d***)

Yield 69% (1.45 g, 3.67 mmol); yellow oil; ^1^H-NMR (500 MHz, CDCl_3_): δ 7.59 (d, 2H, *J* = 7.5 Hz), 7.48 (d, 1H, *J* = 8.0 Hz), 7.43–7.39 (m, 3H), 7.33 (t, 2H, *J* = 7.5 Hz), 7.25 (t, 1H, *J* = 7.5 Hz), 7.20 (t, 1H, *J* = 7.5 Hz), 7.03 (dt, 1H, *J* = 7.5 and 8.0 Hz), 6.96–9.91 (m, 2H), 4.78 (s, 1H), 4.04 (d, 1H, *J* = 6.6 Hz), 4.00 (d, 1H, *J* = 6.6 Hz), 1.93 (s, 1H); ^13^C-NMR (125 MHz, CDCl_3_): δ 162.3 (d, *J* = 247 Hz), 140.0, 138.8, 133.5 (2C, d, *J* = 8.8 Hz), 132.7, 130.4, 128.6, 128.5 (2C), 127.8, 127.6 (2C), 127.4, 124.1, 119.1, 115.5 (2C, d, *J* = 21.6 Hz), 88.7, 84.7, 53.9, 51.2; HRMS (EI) *m*/*z* calcd for C_22_H_17_NBrF [M]^+^ 393.0528, found 393.0524.

### 3.6. N-(2-Bromobenzyl)-3-(4-methoxyphenyl)-1-phenylprop-2-yn-1-amine (***1e***)

Yield 94% (1.00 g, 2.49 mmol); yellow oil; ^1^H-NMR (500 MHz, CDCl_3_): δ 7.61 (d, 2H, *J* = 7.0 Hz), 7.51 (d, 1H, *J* = 8.0 Hz), 7.45 (d, 1H, *J* = 8.0 Hz), 7.42 (d, 2H, *J* = 9.0 Hz), 7.33 (t, 2H, *J* = 7.0 Hz), 7.28–7.22 (m, 2H), 7.07 (t, 1H, *J* = 8.0 Hz), 6.81 (d, 2H, *J* = 9.0 Hz), 4.80 (s, 1H), 4.06 (d, 1H, *J* = 6.6 Hz), 4.04 (d, 1H, *J* = 6.6 Hz), 3.73 (s, 3H)1.92 (brs, 1H); 159.5, 140.3, 138.9, 133.1 (2C), 132.8, 130.5, 128.7, 128.5 (2C), 127.8, 127.7 (2C), 127.4, 124.1, 115.2, 113.9 (2C), 87.5, 85.8, 55.2, 54.0, 51.2; HRMS (EI) *m*/*z* calcd for C_23_H_20_NOBr [M]^+^ 405.0728, found 405.0725.

### 3.7. General Procedure for the One-pot Synthesis of Trisubstituted Allenamide in Figure 2: Synthesis of Allenamide ***2a***

To a solution of propargylamine **1a** (314 mg, 0.835 mmol) in toluene (7 mL), Ac_2_O (0.40 mL, 4.18 mmol) and DBU (0.62 mL, 4.18 mmol) were added at 0 °C under an argon atmosphere. The reaction mixture was stirred at same temperature for 24 h. The reaction was quenched with 1 M HCl. The aq. mixture was extracted with AcOEt. The organic layer was washed with brine, dried over MgSO_4_, filtered, and concentrated in vacuo to give a crude product, which was purified by means of silica gel column chromatography (hexane/AcOEt = 8/1) to afford the allnenamide **2a** (349 mg, 0.834 mmol, 99%).

### 3.8. N-(2-Bromobenzyl)-N-(1,3-diphenylpropa-1,2-dien-1-yl)acetamide (***2a***)

Yield 99% (349 mg, 0.834 mmol); yellow oil; ^1^H-NMR (500 MHz, CDCl_3_): δ 7.46 (d, 1H, *J* = 7.9 Hz), 7.40–7.31 (m, 4H), 7.26–7.20 (m, 5H), 7.07 (t, 1H, *J* = 7.6 Hz), 7.03–7.00 (m, 3H), 6.62 (s, 1H), 5.23 (d, 1H, *J* = 15.3 Hz), 4.72 (d, 1H, *J* = 15.3 Hz), 2.23 (s, 3H); ^13^C-NMR (125 MHz, CDCl_3_): δ 206.9, 171.5, 136.3, 132.7, 132.6, 131.9, 130.0, 129.1 (2C), 128.8 (2C), 128.7, 128.6, 128.3, 127.7 (2C), 127.5, 125.5 (2C), 123.8, 115.8, 101.9, 49.6, 22.2; HRMS (EI) *m*/*z* calcd for C_24_H_20_BrNO [M]^+^ 417.0728, found 417.0730.

### 3.9. N-(2-Bromobenzyl)-N-(1-(4-fluorophenyl)-3-phenylpropa-1,2-dien-1-yl)acetamide (***2b***)

Yield 75% (220 mg, 0.500 mmol); yellow oil; ^1^H-NMR (500 MHz, CDCl_3_): δ 7.46 (d, 1H, *J* = 7.9 Hz), 7.32–7.29 (m, 2H), 7.26–7.22 (m, 4H), 7.09–6.98 (m, 6H), 6.61 (s, 1H), 5.23 (d, 1H, *J* = 15.5 Hz), 4.68 (d, 1H, *J* = 15.5 Hz), 2.24 (s, 3H); ^13^C-NMR (125 MHz, CDCl_3_): δ 206.6, 171.4, 162.9 (d, *J* = 247 Hz), 136.2, 132.7, 130.2, 128.9 (2C), 128.8, 128.7, 128.5, 128.4, 127.7 (2C), 127.6, 127.4 (2C, d, *J* = 8.8 Hz), 123.9, 116.3 (2C, d, *J* = 21.6 Hz), 115.0, 102.1, 49.4, 22.2; HRMS (EI) *m*/*z* calcd for C_24_H_19_NOBrF [M]^+^ 435.0634, found 435.0632.

### 3.10. N-(1-(Benzo[d][1,3]dioxol-5-yl)-3-phenylpropa-1,2-dien-1-yl)-N-(2-bromobenzyl)acetamide (***2c***)

Yield 59% (145 mg, 0.313 mmol); yellow oil; ^1^H-NMR (500 MHz, CDCl_3_): δ 7.46 (d, 1H, *J* = 8.0 Hz), 7.26–7.21 (m, 4H), 7.08–6.98 (m, 4H), 6.84–6.80 (m, 3H), 6.58 (s, 1H), 5.97 (s, 2H), 5.21 (d, 1H, *J* = 15.5 Hz), 4.71 (d, 1H, *J* = 15.5 Hz), 2.24 (s, 3H); ^13^C-NMR (125 MHz, CDCl_3_): δ 206.4, 171.4, 148.6, 148.2, 136.3, 132.7, 132.0, 130.0, 128.9, 128.8 (2C), 128.3, 127.6 (2C), 127.5, 126.6, 123.8, 119.2, 115.7, 108.8, 106.0, 101.9, 101.5, 49.5, 22.1; HRMS (EI) *m*/*z* calcd for C_25_H_20_NO_3_Br [M]^+^ 461.0627, found 461.0622.

### 3.11. N-(2-Bromobenzyl)-N-(3-(4-fluorophenyl)-1-phenylpropa-1,2-dien-1-yl)acetamide (***2d***)

Yield 88% (259 mg, 0.590 mmol); yellow oil; ^1^H-NMR (500 MHz, CDCl_3_): δ 7.45 (d, 1H, *J* = 8.0 Hz), 7.40–7.37 (m, 2H), 7.34–7.31 (m, 1H), 7.24 (d, 1H, *J* = 8.0 Hz), 7.07–7.00 (m, 2H), 6.95–6.87 (m, 6H), 6.60 (s, 1H), 5.31 (d, 1H, *J* = 15.5 Hz), 4.64 (d, 1H, *J* = 15.5 Hz), 2.24 (s, 3H); ^13^C-NMR (125 MHz, CDCl_3_): δ 206.6, 171.5, 162.6 (d, *J* = 247 Hz), 136.3, 132.7, 132.5, 130.0, 129.3, 129.2 (3C), 128.8 (2C, d, *J* = 8.8 Hz), 128.0, 127.9, 127.6, 125.6 (2C), 123.9, 115.9 (2C, d, *J* = 21.6 Hz), 100.8, 49.5, 22.2; HRMS (EI) *m*/*z* calcd for C_24_H_19_NOBrF [M]^+^ 435.0634, found 435.0638.

### 3.12. N-(2-Bromobenzyl)-N-(3-(4-methoxyphenyl)-1-phenylpropa-1,2-dien-1-yl)acetamide (***2e***)

Yield 84% (275 mg, 0.613 mmol); white solid; mp 123.5–157.2 °C (CHCl_3_); ^1^H-NMR (500 MHz, CDCl_3_): δ 7.45 (d, 1H, *J* = 7.5 Hz), 7.38–7.28 (m, 6H), 7.08–7.00 (m, 2H), 6.93 (d, 2H, *J* = 9.0 Hz), 6.75 (d, 2H, *J* = 9.0 Hz), 6.59 (s, 1H), 5.23 (d, 1H, *J* = 15.5 Hz), 4.69 (d, 1H, *J* = 15.5 Hz), 3.78 (s, 3H), 2.25 (s, 3H); ^13^C-NMR (125 MHz, CDCl_3_): δ 206.1, 171.6, 159.7, 136.3, 132.9, 132.6, 129.9, 129.1 (2C), 128.9 (2C), 128.7, 128.5, 127.5, 125.5 (2C), 124.1, 123.8, 115.5, 114.3 (2C), 101.4, 55.4, 49.6, 22.2; HRMS (EI) *m*/*z* calcd for C_25_H_22_NO_2_Br [M]^+^ 447.0834, found 447.0830.

### 3.13. General Procedure for the Palladium-Catalyzed Cascade Reaction of Allenamide with Arylboronic Acid: Synthesis of 1,2-dihydroisoquinoline ***4aa***

To a stirred solution of allenamide **2a** (60.1 mg, 0.144 mmol) in 1,4-dioxane (2.4 mL) and H_2_O (0.6 mL), phenylboronic acid (**3a**) (26.3 mg, 0.216 mmol), Pd(OAc)_2_ (3.2 mg, 0.0144 mmol), P(*o*-tolyl)_3_ (8.7 mg, 0.0287 mmol), and NaOH (28.8 mg, 0.720 mmol) were added at rt under an argon atmosphere. The reaction mixture was stirred for 3 h at 80 °C. Water was added to the reaction mixture, which was extracted with AcOEt. The organic layer was washed with brine, dried over MgSO_4_, filtered, and concentrated in vacuo to give a crude product, which was purified by means of silica gel column chromatography (hexane/AcOEt = 7/1) to afford the 1,2-dihydroisoquinoline **4aa** (53.2 mg, 0.128 mmol, 88%).

### 3.14. 1-(4-Benzhydryl-3-phenylisoquinolin-2(1H)-yl)ethan-1-one (***4aa***)

Yield 88% (53.2 mg, 0.128 mmol); colorless oil; ^1^H-NMR (500 MHz, CDCl_3_): δ 7.33–7.31 (m, 5H), 7.26–7.15 (m, 12H), 7.06 (t, 1H, *J* = 7.5 Hz), 6.94 (t, 1H, *J* = 7.5 Hz), 5.76 (s, 1H), 4.99 (s, 2H), 1.56 (s, 3H); ^13^C-NMR (125 MHz, CDCl_3_): δ 171.3, 142.6 (2C), 138.5, 137.7, 135.3, 132.2, 129.7, 129.4 (3C), 129.3, 129.0, 128.9, 128.7, 128.4, 128.3 (3C), 127.6, 127.2, 126.5 (3C), 126.4, 125.1, 51.4, 46.5, 24.4; HRMS (EI) *m*/*z* calcd for C_30_H_25_NO [M]^+^ 415.1936, found 415.1935.

### 3.15. 1-(4-((4-Methoxyphenyl)(phenyl)methyl)-3-phenylisoquinolin-2(1H)-yl)ethan-1-one (***4ab***/***4ea***)

Yield 82% (52.9 mg, 0.119 mmol) from **2a** with **3b**, and yield 72% (42.1 mg, 0.095 mmol) from **2e** with **3a**; colorless oil; ^1^H-NMR (500 MHz, CDCl_3_): δ 7.35–7.28 (m, 6H), 7.24–7.20 (m, 4H), 7.18–7.14 (m, 4H), 7.06 (t, 1H, *J* = 7.5 Hz), 6.95 (t, 1H, *J* = 7.5 Hz), 6.80 (d, 2H, *J* = 8.5 Hz), 5.71 (s, 1H), 5.02–4.93 (m, 2H), 3.78 (s, 3H), 1.51 (s, 3H); ^13^C-NMR (125 MHz, CDCl_3_): δ 171.3, 158.1, 142.9, 138.3, 137.7, 135.3, 134.5, 132.3, 130.4 (2C), 129.7, 129.3 (3C), 129.2, 128.8, 128.7, 128.3 (3C), 127.6, 127.2, 126.4, 126.3, 125.1, 113.7 (2C), 55.3, 50.6, 46.5, 24.4; HRMS (EI) *m*/*z* calcd for C_31_H_27_NO_2_ [M]^+^ 445.2042, found 445.2041.

### 3.16. 1-(4-((3,5-Dimethoxyphenyl)(phenyl)methyl)-3-phenylisoquinolin-2(1H)-yl)ethan-1-one (***4ac***)

Yield 98% (66.9 mg, 0.141 mmol); colorless oil; ^1^H-NMR (500 MHz, CDCl_3_): δ 7.36–7.31 (m, 5H), 7.27–7.22 (m, 5H), 7.18–7.15 (m, 2H), 7.07 (t, 1H, *J* = 7.5 Hz), 6.97 (t, 1H, *J* = 7.5 Hz), 6.42 (s, 2H), 6.31 (s, 1H), 5.68 (s, 1H), 5.03 (d, 1H, *J* = 13.5 Hz), 4.93 (d, 1H, *J* = 13.5 Hz), 3.68 (s, 6H), 1.56 (s, 3H); ^13^C-NMR (125 MHz, CDCl_3_): δ 171.3, 160.7, 145.0, 142.3, 138.5, 137.7, 135.3, 132.2, 129.7 (2C), 129.4 (2C), 129.0, 128.9, 128.7 (2C), 128.3 (2C), 127.6 (2C), 127.2, 126.4 (2C), 125.0, 107.9 (2C), 98.1, 55.3 (2C), 51.5, 46.5, 24.4; HRMS (EI) *m*/*z* calcd for C_32_H_29_NO_3_ [M]^+^ 475.2147, found 475.2148.

### 3.17. 1-(4-((3,4-Dimethoxyphenyl)(phenyl)methyl)-3-phenylisoquinolin-2(1H)-yl)ethan-1-one (***4ad***)

Yield 86% (57.2 mg, 0.120 mmol); colorless oil; ^1^H-NMR (500 MHz, CDCl_3_): δ 7.36–7.30 (m, 4H), 7.26–7.21 (m, 7H), 7.18–7.14 (m, 1H), 7.07 (t, 1H, *J* = 8.0 Hz), 6.96 (t, 1H, *J* = 7.5 Hz), 6.83 (d, 1H, *J* = 8.5 Hz), 6.78 (d, 1H, *J* = 8.5 Hz), 6.73 (s, 1H), 5.69 (s, 1H), 5.04 (d, 1H, *J* = 13.5 Hz), 4.90 (d, 1H, *J* = 13.5 Hz), 3.86 (s, 3H), 3.69 (s, 3H), 1.55 (s, 3H); ^13^C-NMR (125 MHz, CDCl_3_): δ 171.3, 148.7, 147.6, 142.8, 138.3, 137.7, 135.4, 134.9, 132.3, 129.6 (2C), 129.2 (2C), 128.9, 128.7 (2C), 128.3 (2C), 127.6 (2C), 127.2, 126.5, 126.4, 125.1, 121.7, 112.8, 110.9, 56.0, 55.9, 51.0, 46.5, 24.4; HRMS (EI) *m*/*z* calcd for C_32_H_29_NO_3_ [M]^+^ 475.2147, found 475.2148.

### 3.18. 1-(4-((4-(tert-Butyl)phenyl)(phenyl)methyl)-3-phenylisoquinolin-2(1H)-yl)ethan-1-one (***4ae***)

Yield 92% (63.3 mg, 0.134 mmol); colorless oil; ^1^H-NMR (500 MHz, CDCl_3_): δ 7.34–7.30 (m, 5H), 7.28–7.24 (m, 2H), 7.22–7.20 (m, 5H), 7.16 (d, 4H, *J* = 8.5 Hz), 7.06 (t, 1H, *J* = 7.5 Hz), 6.95 (t, 1H, *J* = 7.5 Hz), 5.73 (s, 1H), 5.09 (d, 1H, *J* = 14.0 Hz), 4.88 (d, 1H, *J* = 14.0 Hz), 1.52 (s, 3H), 1.30 (s, 9H); ^13^C-NMR (125 MHz, CDCl_3_): δ 171.3, 149.2, 142.9, 139.2, 138.3, 137.8, 135.3, 132.3, 129.7 (2C), 129.3 (2C), 129.2, 129.0 (2C), 128.8, 128.7 (2C), 128.2 (2C), 127.7, 127.1, 126.5, 126.3, 125.2 (2C), 125.0, 50.9, 46.5, 34.5, 31.5 (3C), 24.4; HRMS (EI) *m*/*z* calcd for C_34_H_33_NO [M]^+^ 471.2562, found 471.2559.

### 3.19. 1-(4-((4-Chlorophenyl)(phenyl)methyl)-3-phenylisoquinolin-2(1H)-yl)ethan-1-one (***4af***)

Yield 96% (62.0 mg, 0.138 mmol); white solid; mp 205.5–233.9 °C (CHCl_3_); ^1^H-NMR (500 MHz, CDCl_3_): δ 7.33–7.31 (m, 5H), 7.27–7.20 (m, 7H), 7.17–7.14 (m, 4H), 7.08 (t, 1H, *J* = 8.0 Hz), 6.96 (t, 1H, *J* = 8.0 Hz), 5.71 (s, 1H), 5.05 (d, 1H, *J* = 14.0 Hz), 4.90 (d, 1H, *J* = 14.0 Hz), 1.50 (s, 3H); ^13^C-NMR (125 MHz, CDCl_3_): δ 171.26, 142.1, 141.2, 137.5, 135.3, 132.2, 131.9, 130.7 (2C), 129.6, 129.2 (2C), 129.0, 128.8, 128.6 (2C), 128.5 (2C), 128.4, 127.4, 127.3 (2C), 126.7 (2C), 126.5, 125.2, 50.9, 46.5, 24.4; HRMS (EI) *m*/*z* calcd for C_30_H_24_NOCl [M]^+^ 449.1546, found 449.1546.

### 3.20. 1-(4-((4-Fluorophenyl)(phenyl)methyl)-3-phenylisoquinolin-2(1H)-yl)ethan-1-one (***4ag***/***4da***)

Yield 81% (50.8 mg, 0.117 mmol) from **2a** with **3g**, and yield 66% (46.2 mg, 0.107 mmol) from **2d** with **3a**; white solid; mp 154.9–200.0 °C (CHCl_3_); ^1^H-NMR (500 MHz, CDCl_3_): δ 7.35–7.31 (m, 5H), 7.28–7.24 (m, 3H), 7.22–7.16 (m, 5H), 7.07 (t, 1H, *J* = 7.5 Hz), 6.98–6.96 (m, 4H), 5.72 (s, 1H), 5.07 (d, 1H, *J* = 14.0 Hz), 4.89 (d, 1H, *J* = 14.0 Hz), 1.51 (s, 3H); ^13^C-NMR (125 MHz, CDCl_3_): δ 171.2, 161.3 (d, *J* = 244 Hz), 142.4, 138.6, 138.3, 137.6, 135.3, 132.0, 130.8 (2C, d, *J* = 7.9 Hz), 129.6, 129.2 (2C), 129.0, 128.8, 128.7, 128.5 (2C), 127.4 (2C), 127.3, 126.6 (2C), 126.5, 125.2, 115.1 (2C, d, *J* = 21.7 Hz), 50.7, 46.5, 24.4; HRMS (EI) *m*/*z* calcd for C_30_H_24_NOF [M]^+^ 433.1842, found 433.1846.

### 3.21. 1-(4-((2-Acetyl-3-phenyl-1,2-dihydroisoquinolin-4-yl)(phenyl)methyl)phenyl)ethan-1-one (***4ah***)

Yield 85% (55.5 mg, 0.121 mmol); white solid; mp 205.9–220.7 °C (CHCl_3_); ^1^H-NMR (500 MHz, CDCl_3_): δ 7.83 (d, 3H, *J* = 8.5 Hz), 7.33–7.29 (m, 11H), 7.16 (t, 2H, *J* = 7.0 Hz), 7.07 (t, 1H, *J* = 7.5 Hz), 6.95 (t, 1H, *J* = 7.5 Hz), 5.79 (s, 1H), 5.10 (d, 1H, *J* = 14.0 Hz), 4.88 (d, 1H, *J* = 14.0 Hz), 2.55 (s, 3H), 1.51 (s, 3H); ^13^C-NMR (125 MHz, CDCl_3_): δ 197.8, 171.2, 148.4, 141.7, 139.0, 137.5, 135.5, 135.3, 131.9, 129.7, 129.6 (2C), 129.4 (2C), 129.3, 129.2, 129.1, 128.8, 128.6 (2C), 128.4 (2C), 128.2, 127.4, 127.2, 126.8, 126.6, 125.2, 51.5, 46.5, 26.7, 24.4; HRMS (EI) *m*/*z* calcd for C_32_H_27_NO_2_ [M]^+^ 457.2042, found 457.2037.

### 3.22. 1-(4-(Naphthalen-1-yl(phenyl)methyl)-3-phenylisoquinolin-2(1H)-yl)ethan-1-one (***4ai***)

Yield 67% (45.3 mg, 0.0973 mmol); colorless oil; ^1^H-NMR (500 MHz, CDCl_3_): δ 7.81 (d, 2H, *J* = 8.0 Hz), 7.72 (d, 2H, *J* = 8.0 Hz), 7.54–7.51 (m, 4H), 7.41–7.37 (m, 4H), 7.28–7.17 (m, 3H), 7.10 (d, 2H, *J* = 7.0 Hz), 7.01 (t, 2H, *J* = 7.5 Hz), 6.92 (t, 2H, *J* = 7.0 Hz), 6.18 (s, 1H), 5.00 (d, 1H, *J* = 13.0 Hz), 4.89 (d, 1H, *J* = 13.0 Hz), 1.50 (s, 3H); ^13^C-NMR (125 MHz, CDCl_3_): δ 171.3, 143.5, 139.1, 138.3, 137.8, 135.4, 133.9, 132.5, 132.1, 129.7, 129.6 (2C), 129.5, 129.0, 128.9 (2C), 128.7 (2C), 128.5, 128.4, 127.9, 127.8, 127.1, 126.4 (2C), 125.8, 125.5, 125.2 (2C), 125.0, 124.5, 49.5, 46.4, 24.3; HRMS (EI) *m*/*z* calcd for C_34_H_27_NO [M]^+^ 465.2093, found 465.2096

### 3.23. 1-(4-Benzhydryl-3-(4-fluorophenyl)isoquinolin-2(1H)-yl)ethan-1-one (***4ba***)

Yield 55% (42.2 mg, 0.0973 mmol); white solid; mp 211.4–229.1 °C (CHCl_3_); ^1^H-NMR (500 MHz, CDCl_3_): δ 7.32–7.29 (m, 2H), 7.27–7.22 (m, 9H), 7.20–7.14 (m, 3H), 7.06 (t, 1H, *J* = 7.5 Hz), 7.00 (t, 2H, *J* = 8.0 Hz), 6.95 (t, 1H, *J* = 7.5 Hz), 5.69 (s, 1H), 4.97 (s, 2H), 1.53 (s, 3H); ^13^C-NMR (125 MHz, CDCl_3_): δ 171.2, 162.8 (d, *J* = 250 Hz), 142.4 (2C), 137.4, 135.3, 133.7, 132.1, 131.5 (2C), 129.3 (3C), 129.2 (2C, d, *J* = 7.9 Hz), 128.4 (3C), 127.6 (2C), 127.3, 126.6 (3C), 125.1, 115.9 (2C, d, *J* = 21.6 Hz), 51.4, 46.5, 24.5; HRMS (EI) *m*/*z* calcd for C_30_H_24_NOF [M]^+^ 433.1842, found 433.1843.

### 3.24. 1-(4-Benzhydryl-3-(benzo[d][1,3]dioxol-5-yl)isoquinolin-2(1H)-yl)ethan-1-one (***4ca***)

Yield 80% (67.1 mg, 0.146 mmol); white solid; mp 183.1–250.2 °C (CHCl_3_); ^1^H-NMR (500 MHz, CDCl_3_): δ 7.26–7.24 (m, 8H), 7.19–7.16 (m, 3H), 7.13 (d, 1H, *J* = 7.0 Hz), 7.03 (t, 1H, *J* = 7.0 Hz), 6.93 (t, 1H, *J* = 7.0 Hz), 6.83–6.80 (m, 2H), 6.71 (d, 1H, *J* = 8.0 Hz), 5.95 (s, 2H), 5.79 (s, 1H), 4.94 (s, 2H), 1.60 (s, 3H); ^13^C-NMR (125 MHz, CDCl_3_): δ 171.4, 148.1, 147.9, 142.5 (2C), 138.0, 135.2, 132.3, 131.7, 131.4 (2C), 129.4 (3C), 128.3 (3C), 127.5 (2C), 127.1, 126.4 (3C), 125.0, 123.8, 109.8, 108.4, 101.5, 51.5, 46.5, 24.4; HRMS (EI) *m*/*z* calcd for C_31_H_25_NO_3_ [M]^+^ 459.1834, found 459.1835.

## 4. Conclusions

The studies described above resulted in the synthesis of substituted 1,2-dihydroisoquinolines through a palladium-catalyzed cascade cyclization–coupling of trisubstituted allenamides containing a bromoaryl moiety with arylboronic acids. Under the optimum reaction conditions, using 10 mol% of Pd(OAc)_2_, 20 mol% of P(*o*-tolyl)_3_, and 5 equivalents of NaOH in dioxane/H_2_O (4/1) at 80 °C, a variety of highly substituted 1,2-dihydroisoquinolines were concisely obtained. Since the allenamides, as reaction substrates, were prepared from readily available propargylamines in one step, this reaction could provide a useful methodology for the synthesis of 1,2-dihydroisoquinoline derivatives. ^1^H-NMR and ^13^C-NMR characterization of all our synthetic compounds supported the identified structures, the details of which can be found in the Appendix A section.

## Data Availability

The original contributions presented in the study are included in the article/Appendix A, further inquiries can be directed to the corresponding author/s.

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
