# Peer review of "Synthesis of Substituted 1,2-Dihydroisoquinolines by Palladium-Catalyzed Cascade Cyclization–Coupling of Trisubstituted Allenamides with Arylboronic Acids"

_molecules, 2024, doi:10.3390/molecules29122917_

Round 1

Reviewer 1 Report

Comments and Suggestions for Authors Rregarding to the manuscript under consideration: "Synthesis of Substituted 1,2-Dihydroisoquinolines by Palladium-Catalyzed  Cascade Cyclization–Coupling of Trisubstituted Allenamides with Arylboronic Acids", in this MS the authors reported on a variety of substituted 1,2-dihydroisoquinolines were concisely obtained using this methodology. The data are important for the relative fields of Materials Science. 1.     The abstract and conclusion sections should be more informative. 2.     In Introduction section: The introduction part is not well organized. It should be highlighted your novelty. 3.     The synthetic part should be be more specific, such as the quantities, volumes, moles (if possible) of each chemical. This is very important for the reproducibility purpose. 4.     Why not give more comparison on the reported work. 5.     Some topic from the refs may be addressed, such as Org. Chem. Front., 2020,7, 3515-3520; J. Org. Chem. 2019, 84, 14627−14635; Org. Chem. Front., 2021, 8, 4554–4559 and Chem. Commun., 2022, 58, 6653–6656. 6.     What is the mechanism for this reaction, would you add some DFT data? Comments on the Quality of English Language

work in detail

Author Response

For response to Reviewer 1:

Thank you for your valuable and useful comments about our manuscript.

According to your suggestion, we revised the manuscript as follows.

Q1.        The abstract and conclusion sections should be more informative.

A1.        According to the reviewer’s suggestion, we rewritten the abstract and conclusion in the manuscript including more information. Please check the highlighted parts in the manuscript.

Q2.     In Introduction section: The introduction part is not well organized. It should be highlighted your novelty.

A2.        According to the reviewer’s suggestion, we revised the introduction of the manuscript including our novelty. Please check the highlighted parts in the manuscript.

Q3.     The synthetic part should be more specific, such as the quantities, volumes, moles (if possible) of each chemical. This is very important for the reproducibility purpose.

A3.        According to the reviewer’s suggestion, we revised the experimental section of the manuscript which include the quantities and moles of all new compounds.

Q4.     Why not give more comparison on the reported work.

A4.        According to the reviewer’s suggestion, we revised the introduction of the manuscript including comparison on the reported work. Please check the highlighted parts in the manuscript.

Q5.     Some topic from the refs may be addressed, such as Org. Chem. Front., 2020,7, 3515-3520; J. Org. Chem. 2019, 84, 14627−14635; Org. Chem. Front., 2021, 8, 4554–4559 and Chem. Commun., 2022, 58, 6653–6656.

A5.        Thank you for providing us about useful information. Although the addressed manuscripts were very valuable, out manuscript cannot cite them because of the low relevance.

Q6.     What is the mechanism for this reaction, would you add some DFT data?

A6.        The reaction mechanism for the palladium-catalyzed cascade cyclization of allenamides were well investigated. Instead of DFT data, we cited from the related manuscripts about the description of reaction mechanism in page5, line 124.

Reviewer 2 Report

Comments and Suggestions for Authors

The presented paper is devoted to synthesis of different substituted 1,2-dihydroisoquinolines using palladium-catalyzed cascade cyclization-coupling of trisubstituted allenamides. The data obtained are novel, important and useful for variety researchers dealing with biological or medical fields or fine organic synthesis. The data is presented clearly and step by step. To confirm all resulted products during proceeding reactions the 1H-NMR and 13C-NMR methods have been applied and all corresponding NMR spectra are provided in supplementary information. The manuscript can be published taking into account the following comments:

1)    In conclusions, the authors say that a variety of substituted 1,2-dihydroisoquinolines has been obtained. Since all products confirmed by NMR presented in SI, I would recommend to slightly extend conclusions by adding a sentence about confirmation of the products by NMR.

2)    Continuing from the previous comment, the authors are kindly asked to add corresponding sentence with references to Figures SI directly in the main text of the manuscript.

3)    Some schemes, figures or tables could be clearer, if the resulting products were colored into different colors, e.g., scheme 2 can be improved by coloring substituted patterns by other colors like presented in table 1.

4)    Page 6, line 138, There is a typo: “chemical shift”, not “sift”.

Author Response

For response to Reviewer 2:

Thank you for your valuable and useful comments about our manuscript.

According to your suggestion, we revised the manuscript as follows.

Q1.  In conclusions, the authors say that a variety of substituted 1,2-dihydroisoquinolines has been obtained. Since all products confirmed by NMR presented in SI, I would recommend to slightly extend conclusions by adding a sentence about confirmation of the products by NMR. 

Q2.  Continuing from the previous comment, the authors are kindly asked to add corresponding sentence with references to Figures SI directly in the main text of the manuscript.

A1, A2. According to the reviewer’s suggestion, we revised the conclusion of the manuscript including the description about NMR confirmation and SI. Please check the highlighted parts in the manuscript.

Q3.  Some schemes, figures or tables could be clearer, if the resulting products were colored into different colors, e.g., scheme 2 can be improved by coloring substituted patterns by other colors like presented in table 1. 

A3. According to the reviewer’s suggestion, we revised the figure of Scheme 2 by coloring substituted patterns.

Q4.  Page 6, line 138, There is a typo: “chemical shift”, not “sift”.

A4.  According to the reviewer’s suggestion, we corrected a manuscript as shown in page 6, line 145.

Reviewer 3 Report

Comments and Suggestions for Authors

This manuscript described an efficient synthetic strategy to substituted isoquinoline derivatives utilizing palladium-catalyzed domino reactions. Although similar strategy has already been used in the synthesis of isoquinolinone by Grigg and coworkers (ref. 32), this work began with more complicated allenamides, which were prepared by authors’s own method (ref. 38) to produce isoquinoline skeleton instead of isoquinolinone. Thus, this method is quite valuable for not only synthetic chemist but also medicinal chemist, who needs various chemical entities to find biologically active compounds. Overall the manuscript proves its own novelty and all substrate scopes of the reactions are well studied to clarify application scope and limitations. Thus, as a reviewer, I recommend the manuscript to be published in Molecules after revision and/or answer to comments in below.

Line 61: In scheme 2, “R = CH3” is not necessary. Please erase five of those.

Question 1: Why do you use bromoaryl substrates as starting material of palladium-catalyzed reaction? Probably, iodoaryl substrates are more reactive toward oxidative addition step, which might increase yield and make reaction conditions mild. I just assume that the reason might come from commercial availability difference between 2-bromobenzylamine vs 2-iodobenzylamine. If you have any other reason, please describe in the text.

Question 2: Is there any reference related to the first conditions (Pd(OAc)2, P(o-tolyl)3, NaOH, dioxane/H2O)?

If you have one, please cite the reference in the text.

Author Response

For response to Reviewer 3:

Thank you for your valuable and useful comments about our manuscript.

According to the reviewer’s suggestion, we revised the manuscript as follows.

Suggestion: Line 61: In scheme 2, “R = CH3” is not necessary. Please erase five of those.

Answer: According to the reviewer’s suggestion, we corrected the description of Scheme 2.

Q1. Why do you use bromoaryl substrates as starting material of palladium-catalyzed reaction? Probably, iodoaryl substrates are more reactive toward oxidative addition step, which might increase yield and make reaction conditions mild. I just assume that the reason might come from commercial availability difference between 2-bromobenzylamine vs 2-iodobenzylamine. If you have any other reason, please describe in the text.

A1.        As the reviewer pointed out, we selected the use of bromoaryl substrates as starting material for the reason of commercially availability of 2-bromobenzylamine. We revised the manuscript including the phrase “commercially available”in page 3, line 68.

Q2. Is there any reference related to the first conditions (Pd(OAc)2, P(o-tolyl)3, NaOH, dioxane/H2O)? If you have one, please cite the reference in the text.

A2.        As the reviewer pointed out, we selected the first conditions by following the standard Miyaura-Suzuki coupling. We cited from the related manuscript [39] in page 4, line 82.